# Whole-of-school physical activity implementation in the context of the Dubai Fitness Challenge

**Chris McMahon**[1], **Collin A. Webster**[2]*, **R. Glenn Weaver**[3], **Christophe El Haber**[4], **Gönül Tekkurşun Demir**[5], **Zainab Mohamed Ismail**[4], **Syeda Zoha Fatima Naqvi**[4], **Mehnaz Ghani**[4], **Şevval Kepenek**[4], **Manel Kherraf**[4], **Thrisha Krishnakumar**[4], **Pranati Prakash**[4], **Yeowon Seo**[4]

**1** University of Birmingham, School of Sport, Exercise and Rehabilitation Sciences, Birmingham, United Kingdom, **2** Department of Kinesiology, Texas A&M University–Corpus Christi, Corpus Christi, TX, United States of America, **3** University of South Carolina, School of Public Health, Columbia, SC, United States of America, **4** University of Birmingham Dubai, School of Psychology, Dubai, UAE, **5** Gazi University, Faculty of Sport Sciences, Ankara, Turkey

* collin.webster@tamucc.edu

## Abstract

### Introduction

Physical activity (PA) promotion among school-aged youth is a global health priority. Recommendations for such promotion include implementing whole-of-school approaches that maximize resources across the school environment. This study examined schools' participation in an annual, government-led, and emirate-wide initiative in Dubai, called the Dubai Fitness Challenge, in which the goal is to accrue 30 minutes of PA every day for 30 days (as such, the initiative is colloquially referred to as "Dubai 30x30").

### Methods

A mixed-methods design was employed for this study. Three schools were recruited using convenience sampling. Participants were 18 physical education teachers, 20 classroom teachers, 2 principals and 45 students. Data sources included surveys, focus groups, and interviews. Data were analyzed using descriptive statistics, multinomial logistic regression, and open and axial coding to develop themes.

### Results

School staff reported that most Dubai 30x30 activities were provided in physical education, at break times during school, and before and after school. Students reported that they mainly participated in Dubai 30x30 activities during physical education and occasionally participated in activities after school and on weekends. During school, students were more likely to reach higher PA intensity levels when they were in contexts other than the regular classroom setting. Among school staff, physical education teachers were most involved and classroom teachers were least involved in promoting Dubai 30x30. Parent engagement was high. Staff perceived that Dubai 30x30 brought the community together, but physical

**Data Availability Statement:** The raw data for this study are confidential as they focus on children and school professionals who participated in surveys, observations, interviews and focus groups. We

have removed all identifying information from our data sets and have uploaded these as supporting files with our manuscript.

**Funding:** The authors received no specific funding for this work.

**Competing interests:** The authors have declared that no competing interests exist.

education teachers also indicated there was a lack of implementation guidance and they felt burdened. Participants believed Dubai 30x30 increased PA participation and helped to promote their schools.

## Discussion

This study provides an initial glimpse into schools' participation in Dubai 30x30 and suggests that a whole-of-school PA lens is useful in gleaning information that could help to increase and optimize PA opportunities for students.

## Introduction

Physical activity (PA) benefits children and adolescents by supporting their physical, mental, social, and emotional development [1]. The World Health Organization (WHO) recommends that school-age youth engage in at least 60 minutes of moderate-to-vigorous intensity PA each day [2]. This PA should be mostly aerobic and should include activities that strengthen muscles and bones at least three days per week [2]. Unfortunately, 81% of adolescents do not meet PA guidelines [3]. As physical inactivity can track from adolescence into young adulthood [4,5], early intervention during the school years should be a priority for addressing insufficient PA levels in youth and promoting active living.

Given their extensive reach and existing resources, schools provide an attractive setting for supporting children and adolescents' PA needs. Recommendations stress the importance of taking a "whole-of-school approach" to PA promotion [6–8]. Such an approach may look different from one school to the next, but the general idea is to maximize PA opportunities and participation by using as many resources as possible before, during and after school. For example, the comprehensive school physical activity program (CSPAP) framework, which is the national framework for school-based physical education and PA in the United States, identifies five components that can be used synergistically to help ensure all school-aged youth meet the PA guidelines and develop the competence and confidence to lead physically active lives [9]. These components include (a) quality physical education, (b) PA during school, (c) PA before and after school, (d) staff involvement and (e) family and community engagement. A quality physical education program should serve a foundational role in a CSPAP by providing students with the knowledge and skills to pursue a physically active lifestyle. Other PA opportunities during school (e.g., recess and other break times, PA integrated with classroom time), as well as PA opportunities before and after school (e.g., clubs, active transportation), can allow students to apply and extend what they learn in physical education. The staff involvement and family and community engagement components focus on recruiting the support of as many other people as possible (e.g., all teachers, school administrators, parents, community partners) to assist with PA promotion.

Evidence suggests that a whole-of-school approach to PA promotion leads to positive outcomes for school-age youth [10–13]. For example, Pulling Kuhn et al. conducted a systematic review of multicomponent school-based PA interventions and found that reported outcomes included increased time spent in PA, increased physical fitness, improved motor skills, improved weight status, improved blood pressure, improved on-task behavior and improved performance in reading and math [13]. However, surveys conducted in the U.S. indicate whole-of-school PA programs exist in only 16% of elementary schools and 1–7% of secondary schools [11,14,15]. In Ireland, a national initiative with primary (elementary) schools called

Active School Flag has had more success with 85% of schools engaging at some level with the initiative and 46% of schools earning at least one flag (i.e., meeting criteria to be named an "active school") [10].

The varying uptake of whole-of-school PA underscores the importance of examining schools' implementation of current initiatives. Carson et al. [16] proposed a conceptual framework for research and practice related to whole-of-school PA, which is based on ecological systems theory [17,18]. The framework is organized into multiple levels of influence on the implementation of a whole-of-school PA initiative and, ultimately, on youth PA. Specific to implementation, the most proximal level of influence is the Meso level. This level consists of implementation facilitators, including the knowledge, skills, and dispositions of the implementers (e.g., school personnel), as well as resources (e.g., school facilities, equipment, time, and space allocation) and safety (physical, social, and emotional) regarding PA opportunities. Above the Meso level is the Exo level, which comprises the leaders of whole-of-school PA initiatives: a champion (someone who spearheads the initiative), the school administrative team (i.e., their support for the initiative) and a school committee dedicated to the initiative. Finally, the Macro level incorporates cultural elements of the school in relation to PA promotion. These elements include policy (e.g., PA-related legislation, monitoring and evaluation of the initiative) and normative behaviors and beliefs (e.g., shared expectations and values of members of the school community).

Research on whole-of-school PA implementation has identified numerous factors that align with all levels of influence in Carson et al.'s [16] conceptual framework. These factors include public policies (e.g., state mandates related to whole-of-school PA) [19], school culture (e.g., prioritizing PA amid the pressures of academic testing, PA role-modelling by school staff, celebrating PA participation) [20,21], program leadership (e.g., a person who champions the initiative) [22], administrative support (e.g., providing teachers with PA promotion resources, allowing for teacher autonomy in program implementation, administrators role-modelling PA) [19,20], facilities (e.g., physical education facilities, bike racks that support active commuting) [19], knowledge and beliefs related to PA promotion [20] and the availability of many PA opportunities [22]. Such research is critical to informing ongoing and future efforts to maximize the implementation of whole-of-school PA initiatives.

In parallel with global trends, 81% of children and adolescents in the United Arab Emirates are not accumulating at least 60 minutes of moderate-to-vigorous PA each day [23]. One initiative that may help to address this issue is the Dubai Fitness Challenge. Since 2017, the Dubai government has implemented the Dubai Fitness Challenge (also referred to as Dubai 30x30) annually with the goal of engaging Dubai residents in 30 minutes of fitness activities for 30 days (late October through late November). Numerous activities are organized throughout the month, such as a half marathon, a bicycle ride, sports competitions, video workouts and free fitness classes. According to the Khaleej Times (citing data from Dubai's Department of Economy and Tourism) there were 786,000 participants in the first year of the event's implementation and 1.6 million participants in 2021 (close to half of Dubai's total population) and 88% of those who participated in 2021 reached the 30/30 goal.

Though not a requirement, Dubai schools are strongly encouraged to participate in Dubai 30x30. Some media attention has been given to reporting various kinds of activities that schools implement during the initiative and the benefits of the initiative according to physical education teachers [24–26]. Examples of activities include partnering with community organizations/companies to offer different sports, games, and fitness events, taking field trips (e.g., to the beach) to exercise, having students use activity passports or scorecards to incentivize participation, and engaging school staff and parents in activity participation. Reported benefits include students wanting to continue doing activities they were introduced to throughout the

month and parents' enjoying participation in the initiative. Despite these media portrayals of Dubai 30x30 implementation in schools, there is a lack of research that examines such implementation. No studies have investigated schools' implementation of the initiative from the perspective of whole-of-school PA promotion, which is essential for helping school leaders (e.g., government officials who oversee the education sector in Dubai, school administrators) optimize Dubai 30x30 implementation using available school and community resources. The purpose of this study, therefore, was to examine schools' implementation of Dubai 30x30 through a whole-of-school PA lens. As this was the first study to investigate schools' implementation of Dubai 30x30, we opted to approach the study in a broad and exploratory manner by asking the following research questions, which consider the multiple tiers, contexts, and factors that comprise whole-of-school PA frameworks [7,8,16,27]: (a) What was the extent and nature of Dubai 30x30 activities for students, staff, and families before, during and after school? (b) To what extent did students participate in Dubai 30x30 activities? (c) How did students' PA levels during school relate to contextual variables? (d) To what extent and in what ways were staff, families and community partners involved/engaged in promoting Dubai 30x30? (e) What were the perceived successes and challenges involved with implementing the initiative? and (f) What was the perceived impact of implementing the initiative on students, staff, families, and the community?

## Materials and methods

### Research design

This descriptive study was conducted using a convergent parallel mixed-methods research design [28]. Specifically, quantitative and qualitative data were collected concurrently and analyzed both separately and jointly to address the research questions. A convergent parallel design suited our research because it (a) allowed us to develop a more complete understanding of the research phenomenon than what either quantitative or qualitative methods could produce alone, (b) increased the efficiency of data collection within the relatively short timeframe of Dubai 30x30, and (a) permitted us to capitalize on the strengths of the multiple researchers on our team who have backgrounds in quantitative and qualitative science [29]. Within this research approach, the conceptual framing for whole-of-school PA guided data collection, as such framing highlights the importance of investigating PA opportunities and their outcomes as part of a system that consists of multiple interacting layers, contexts, and factors [7,8,16,27]. As this study was descriptive in nature, our intention was not to evaluate schools' implementation of Dubai 30x30 or determine changes across time (e.g., via an experimental design), but rather to document processes and outcomes that characterized such implementation, including elements that were both unique to Dubai 30x30 and part of schools' routine practices that support a physically active school environment.

### Participants and setting

Convenience sampling was used to recruit three private international schools in Dubai to participate in the study. Although, as previously mentioned, it has been reported that participation in Dubai 30x30 is high, gaining access to schools for research can be challenging. We drew upon already established school partnerships in Dubai and invited five schools to participate but two declined. Each participant school follows a British curriculum. The first school (School 1) serves students ages 3–11 and the second and third schools (School 2 and School 3) serve students ages 3–18. School 1 opened in 1989, School 2 opened in 2015 and School 3 opened in 2021. At the time of this study, more information was available for Schools 1 and 2 than for School 3 because the first two schools had undergone government inspections and reports of

the inspection results were publicly available online, whereas the third school had not yet been inspected. School 1 had a total enrollment of 1177 students and a teacher-student ratio of 1:15. School 2 had a total enrollment of 1020 students and a teacher-student ratio of 1:8. School 3 had a total enrollment of 1290 students but there was no information available about teacher-student ratio. The largest nationality group of students at School 1 was UK, while the largest nationality group of students for School 2 was European. There was no information available about students' nationalities for School 3. For the 2022–2023 school year, the Dubai government rated School 1 as "outstanding" and School 2 as "very good." These ratings encompass numerous areas of focus, such as students' achievement, personal and social development, teaching and assessment, staffing, facilities, and other resources. While School 3 had not yet received a rating, our visits to the school led us to believe that the learning climate and availability of resources were similar in quality compared to the first two schools. Compared to the public schools in Dubai, private schools tend to be rated higher and have better resources. Overall, about 90% of schools in Dubai are private schools.

Each participant school was notified that all students, teachers, administrators, and parents were eligible to participate in the study. However, as participation was voluntary and there was no tangible incentive for participation, the final number of participants was smaller than we hoped for. Participant details from each school are displayed in Table 1. There were four participant groups: students, physical education teachers, classroom teachers and principals. Our

**Table 1. Participant details.**

|  | N | F | M | $M_{Age}$ (SD) |
|---|---|---|---|---|
| Surveys |  |  |  |  |
| Students | 45 | 64% | 24% | 12.22 (1.47) |
| Physical Education Teachers | 18 | 39% | 49% | 34.92 (3.94) |
| Classroom Teachers | 20 | 75% | 10% | 36.06 (5.72) |
| Principals | 2 | 100% | 0% | NR |
| Observations |  |  |  |  |
| Observation 1 (Year 6 Class) | 11 | 55% | 45% | NR |
| Observation 2 (Year 1 Class) | 2 | 100% | 0% | NR |
| Observation 3 (Year 3 Class) | 14 | 36% | 64% | NR |
| Observation 4 (Year 4 Class) | 18 | 56% | 44% | NR |
| Observation 5 (Year 4 Class) | 17 | 35% | 65% | NR |
| Observation 6 (Year 5 Class) | 23 | 74% | 26% | NR |
| Observation 7 (NR) | 2 | 100% | 0% | NR |
| Observation 8 (Year 6 Class) | 9 | 67% | 33% | NR |
| Observation 9 (Year 4 Class) | 13 | 62% | 38% | NR |
| Observation 10 (Year 5 Class) | 2 | 100% | 0% | NR |
| Observation 11 (Year 4 Class) | 25 | 36% | 64% | NR |
| Observation 12 (Year 3 Class) | 4 | 75% | 25% | NR |
| Observation 13 (Year 7 Class) | 17 | 88% | 12% | NR |
| Observation 14 (Year 1 Class) | 6 | 50% | 50% | NR |
| Observation 15 (Year 5 Class) | 4 | 75% | 25% | NR |
| Focus Groups and Interviews |  |  |  |  |
| Physical Education Teachers | 17 | 22% | 78% | 36.06 (6.18) |
| Classroom Teachers | 5 | 80% | 20% | 35.00 (7.35) |
| Students | 7 | 57% | 43% | 13.25 (0.71) |

Note: NR = Not reported.

aim was to understand the perspectives of as many different participant groups as possible, given that whole-of-school PA implementation calls for the involvement and engagement of the whole school community [7,8,27]. While we sought representation from parents, no parents volunteered to participate in the study. Participation is reported in terms of the number of individuals from each participant group who participated in each method of data collection (survey, observations, and focus groups/interviews; see the following section on data sources and procedures). In all, there were 78 survey participants, 15 classes that were observed and 29 focus group/interview participants. Age bands for the different school years (grades) are 5–6 for Year 1, 6–7 for Year 2, 7–8 for Year 3, 8–9 for Year 4, 9–10 for Year 5, 10–11 for Year 6 and 11–12 for Year 7.

## Data sources and procedures

The Ethics Board at the University of School-Based Science approved this study.

School and participant recruitment was conducted from August 8th, 2022, to January 30th, 2023. Participants provided written informed consent/assent prior to the start of data collection. School professionals provided their consent, parents/guardians provided consent for student participants, and students provided their assent. Consent/assent forms explained the benefits and risks of participating in the study and the procedures that would be used to maintain confidentiality of the data. Data sources for this study included surveys, school observations, focus groups and individual interviews. The surveys and school observations provided quantitative data, while the focus groups and individual interviews provided qualitative data. A physical education teacher at each school served as the point of contact during the study and helped to recruit participants and facilitate data collection.

**Surveys.** Schools 2 and 3 participated in electronic surveys that were administered over a three-week period directly following Dubai 30x30 (School 1 opted not to participate in the surveys). The online survey platform JISC was used to create and distribute the surveys and manage the data. Weekly reminders were sent via email to increase the number of survey respondents. There were four surveys: one for students, one for physical education teachers, one for classroom teachers and one for principals. Students completed their survey during normal classroom time and with assistance from teachers, as needed. We encouraged teachers to administer the survey during periods of the school day that were least disruptive to teaching and learning and to assist students only by providing logistical support (e.g., using a computer to take the survey) and clarification if students did not understand the meaning of certain questions or terminology used in the survey, but not to suggest to students how to respond to any of the questions.

Survey items pertinent to the current study focused on Dubai 30x30 activities before, during and after school (Research Question 1), students' participation in these activities (Research Question 2) and school staff's involvement in Dubai 30x30 (Research Question 4). Regarding Dubai 30x30 activities, school staff were asked to indicate the extent to which activities were provided in each of six contexts (physical education, classroom, recess/break, before school, after school, and weekends), whereas students were asked to indicate their level of participation in these activities. A six-point response scale was used. The response scale for school professionals ranged from "No Dubai 30x30 activities were provided" to "A lot of Dubai 30x30 activities were provided." Participants could also choose an option that stated, "I don't know where there were any Dubai 30x30 activities provided." The response scale for students ranged from "I did not participate at all in Dubai 30x30 activities" to "I participated a lot in Dubai 30x30 activities." Students could also choose the option, "There were no Dubai 30x30 activities provided."

Items assessing staff involvement were adapted from existing measures developed in previous whole-of-school research [30–32]. Example items include "I am involved with Dubai 30x30 planning at my school" (asked to all school staff), "I am involved with integrating Dubai 30x30 activities into physical education lessons" (asked only to physical education teachers), "I am involved with integrating Dubai 30x30 activities into my classroom lessons or as breaks/transitions during normal classroom time" (asked only to classroom teachers) and "I am involved with allocating resources for my school's participation in Dubai 30x30" (asked only to principals). A six-point response scale was used, which ranged from "Strongly Disagree" to "Strongly Agree."

The surveys also included a demographic section at the end. All participants were asked to identify their gender and their race/ethnicity. Additionally, students were asked to identify their age, while school staff were asked to indicate the levels at which they were teaching (e.g., elementary, secondary). An optional response format was used for all items on the surveys so that participants could skip questions if they wished to do so.

**Observations.** The Observational System for Recording Physical Activity in Children–Elementary School (OSRAC-E) [33] was used to examine the relationship between students' PA levels and contextual variables during school (Research Question 3). Although the instrument was designed for elementary school settings, much of the category system can be applied to secondary school settings, as well. Five categories from the instrument were used for the purposes of the present study: activity level, location, physical setting, instructional setting, and activity context. OSRAC-E uses a focal child protocol, whereby a single child is observed at a time using momentary time sampling. We employed a five-second observe interval followed by a 25-second record interval, which was repeated for five minutes with the same child. This observation procedure was used with as many children as possible during each school visit during the study.

In the week before observations were scheduled to commence, the second author conducted a two-hour training session on the observation protocols used for the study. Trainees were undergraduate students who volunteered to help with data collection and other aspects of the research. The session involved familiarizing the trainees with the OSRAC-E and testing the trainees' reliability in using the instrument. Inter-observer reliabilities were calculated using the following formula [34]:

$$\frac{Number\ of\ Agreements}{Number\ of\ Agreements + Disagreements} \times 100 = \%\ Agreement$$

Video examples were used for the training and to test inter-observer reliability. A reliability standard of 80% was considered acceptable [34]. Total of eight trainees demonstrated acceptable inter-observer agreement proceeded with conducting observations the following week.

A second two-hour training session was held after the first week of data collection to help prevent observer drift and address any unanticipated challenges with data collection. During this session, observers shared their experiences doing the observations and suggested ways to help increase inter-observer reliability. Suggestions mainly focused on how to record unusual activities or types of behavior using the OSRAC-E. The session also included additional opportunities for observers to practice using the using more video examples and concluded with another round of reliability testing, which resulted in acceptable levels of agreement (above 80%) for all observers.

Observations were conducted across 15 classrooms from Schools 2 and 3 during the full month of Dubai 30x30 (School 1 opted not to participate in the observations). Observation days and times were scheduled in accordance with when the observers were available and

when the schools were able to accommodate observers. In total, observations were conducted for 10.25 hours across 12 days. During each school visit, observers usually stayed with a single class of students and followed the class during the full observation period (approximately 2–7 hours). This allowed for data to be collected in different instructional contexts (e.g., home-room, physical education). Observation shifts lasted two and a half hours for each observer to help prevent exhaustion and loss of focus. Additional inter-observer reliability checks were conducted in the third and fourth weeks of data collection with agreement scores indicating acceptable reliability.

**Focus groups and interviews.**   A total of six focus groups and two individual interviews were conducted (see Table 1 for participant details). One focus group with physical education teachers and one individual interview with a physical education teacher were conducted at School 1, two focus groups with students were conducted at School 2, and three focus groups (one with physical education teachers and two with classroom teachers) and an individual interview with the principal were conducted at School 3. The purpose of the focus groups and interviews was two-fold. First, they were intended to provide additional information about Dubai 30x30 activities (Research Question 1) and the involvement/engagement of school staff, families, and community partners in promoting Dubai 30x30 (Research Question 4). Second, they were designed to address school staff's perceived successes and challenges involved with implementing Dubai 30x30 (Research Question 5) and participants' perceived impact of the initiative on students, families, and the community (Research Question 6). A semi-structured format was used to allow for some flexibility in each focus group/interview. Examples of questions include "What did your school do in the name of Dubai 30x30 this school year?", "Describe your involvement in Dubai 30x30 this school year as a physical education/classroom teacher/principal. What roles did you play?", "From your perspective, what were notable successes for your school during its involvement with Dubai 30x30 this school year?", "Describe any challenges or problems you remember from your school's involvement with Dubai 30x30 this school year?" and "What was the impact of your school's involvement in Dubai 30x30?" Additionally, prompts were used for clarification and to explore participants' responses in more depth. Following recommended protocols, two researchers conducted each focus group with one researcher asking questions and the other researcher taking notes and helping to moderate, when needed [35]. Interviews/focus groups generally lasted about 30 minutes, were conducted either in person at participants' schools or online via Zoom and were audio recorded.

## Data analysis

**Surveys.**   Microsoft Excel was used to calculate descriptive statistics, including means and standard deviations, for the extent of Dubai 30x30 activities before, during and after school (Research Question 1), the extent of students' participation in Dubai 30x30 activities (Research Question 2) and the extent of involvement/engagement of school staff in promoting Dubai 30x30 (Research Question 4).

**Observations.**   To examine the relationship between students' PA and contextual variables during school (Research Question 3), the likelihood of students engaging in each activity level was estimated using multinomial logistic regression with students engaging in stationary as the referent group. Analyses were completed using STATA (v.16.1, STATA Corporation LLC, College Station, TX).

**Focus groups and interviews.**   All audio recordings were transcribed verbatim for analysis. Open and axial coding were used to develop themes in the data [36]. Specifically, the first author read each transcript to become familiar with participants' responses. Then, he initiated

open coding, which involved writing memos that summarized/interpreted the responses (or parts of responses) in each transcript that seemed helpful in addressing Research Questions 1, 4, 5 and 6. This process continued iteratively, allowing for revisions and additions to the memos as the author made additional passes through the transcripts and developed his understanding of the major ideas underlying the data. Subsequently, the author used axial coding to search across transcripts for memos suggesting similar ideas. He synthesized these memos into broader ideas related to each research question to develop themes. To help ensure trustworthiness [37], the second author served as a peer debriefer throughout the analysis, regularly meeting with the first author to discuss the analysis process, the coding, and the themes. Trustworthiness was also established through the triangulation of multiple data sources.

## Results

### Research Question 1: Dubai 30x30 activities before, during and after school

Descriptive statistics and themes related to Research Question 1 are shown in Table 2. School staff indicated on their surveys that there were more Dubai 30x30 activities provided during physical education than in other contexts. Recess/other breaks during school and before and after school periods were also identified as times when there were a fair amount of Dubai 30x30 activities provided. Participants felt that there were very few Dubai 30x30 activities provided during classroom time and on weekends. Themes and supporting evidence from the focus group and interview data are presented in Table 3. The themes converged with the survey data in that all participant groups (physical education teachers, classroom teachers and students) and the principal spoke about Dubai 30x30 activities that were offered during break times and before and after school. There was also a theme characterized by a recurring competitive element to many of the activities. Surprisingly, physical education was seldom mentioned in relation to Dubai 30x30 activities, which contrasted with the survey data.

### Research Question 2: Students' participation in Dubai 30x30 activities

Table 4 displays the descriptive statistics related to Research Question 2. Consistent with school staff's survey responses regarding the extent of Dubai 30x30 activities provided in different contexts, students reported participating a fair amount in Dubai 30x30 activities during physical education and very little in such activities during classroom time. However, in contrast to the school staff survey data, students indicated they participated very little in Dubai 30x30 activities during recess/other break times and occasionally participated in initiative-related activities after school and on weekends.

Table 2. Research Question 1: Extent of Dubai 30x30 activities provided before, during and after school (combined descriptive data from surveys administered to physical education teachers, classroom teachers, and principals at schools 2 and 3).

|     | Physical Education | Classroom | Recess/Other Break Times | Before School | After School | Weekends |
|-----|------|------|------|------|------|------|
| M   | 4.90 | 2.81 | 4.79 | 4.16 | 4.70 | 2.66 |
| SD  | 1.72 | 1.45 | 1.32 | 1.55 | 1.41 | 1.88 |
| N   | 40   | 32   | 39   | 38   | 37   | 32   |

Note: The response scale ranged from 1 ("No Dubai 30x30 activities were provided") to 6 ("A lot of Dubai 30x30 activities were provided").

**Table 3. Research Question 1: Extent and nature of Dubai 30x30 activities provided before, during and after school (themes and supporting evidence from focus group and interview data from all three schools).**

| Theme | Supporting Evidence |
|---|---|
| Break Time | "During break times we had opportunities for students to take part in organized activities, so we opened the [sports facilities] to the year sevens so they could participate in various activities such as football, basketball, and netball" (PE Teacher 3, School 1)<br>"So, we had, like Miss Stanley would often go out, and [put] the music on [at] lunchtime. We'd have a chance to like, dance and get active" (Classroom Teacher 1, School 3)<br>"We did daily activities with the children. Those were done at break time options, with different Dubai 30 activities" (Principal, School 3)<br>"There were cool [activities]. . .at break times there were many teachers and basketball kinds of stuff" (Student 1, School 2) |
| Before and After School | "Before school, we had the various activities that relate to PE and sport. . .we had swimming. . .there was netball as well, athletics down at the stadium, there was golf also. . .rugby, football, fitness, yeah, the fitness sessions for the staff as well, fitness session for the students, for parents. . .there were swimming activities also" (PE Teacher 3, School 3)<br>"So, our clubs start at 6.45. So, and that takes us up then to registration. So, we [were] there was Monday, Tuesday, Wednesday, Thursday, Friday. There was something on every day and then at 7.30 I would go and do the Wake Up, Shake Ups twice a week as well" (PE Teacher 1, School 1)<br>When asked to discuss some of the activities taking place after school, PE Teacher 1 from School 3 said the following "Netball, basketball, rugby, football, table tennis, tennis, football, skiing, taekwondo, jujitsu, karate, gymnastics, acro dance, musical theatre, ballet and that is most of the activities run from FS all the way through to year 7"<br>"We had a color run thing at school with the parents that involved, and one evening as well" (Classroom Teacher 1, School 3) |
| Competitions | PE Teacher 1 from School 3 who said "So we try and get them involved in a competition format as well" (PE Teacher 1, School 3)<br>"We obviously did a competition as well, so those that did, you know, complete the diaries the most we gave them T-shirts, and we just made a big song and dance of it in assembly" (PE Teacher 1, School 2)<br>"[Students] also get awards within the classroom for [participation] as well. But they're also included within the house reward system. So, house points" (PE Teacher 3, School 3)<br>"We had a full competition with the diary daily. So, the children could, if they took part in that Dubai 30 challenge. . .they recorded it in a journal, and then [submitted] that into the PE Team Department. I can't (remember) the prize what they got for that, house points, I think" (Classrom Teacher 3, School 3)<br>"There was quite a lot of challenges going on as well" (Classroom Teacher 1, School 3)<br>"There was this one dance competition in school between houses like we had to like, choose which house that's the best and it was on the field" (Student 1, School 2)<br>When asked what gets him excited about participating in Dubai 30x30, Student 2's (School 2) immediate response was "Competition" |

## Research Question 3: Relationship between students' PA levels and contextual variables during school

Results of the multinomial logistic regression models are presented in Table 5. Several physical setting variables were related to students' engagement in activity. For example, children were 2.39 (95CI = 1.08, 5.28) and 52.36 (95CI = 13.12, >100.00) times as likely to engage in slow easy and fast activity in the gym compared to the classroom. Further, the playground, sports fields and other physical settings showed similar relationships with children's physical activity. At the playground children were 6.02 (95CI = 2.02, 17.92), 6.48 (95CI = 3.49, 12.02) and 110.66 (95CI = 31.34, >100.00) times as likely to engage in slow easy, moderate, and fast activity compared to the classroom. At the sports field children were 5.69 (95CI = 2.65, 12.02), 4.86 (95CI = 1.85, 12.77) and 55.34 (95CI = 12.52, >100.00) times as likely to engage in slow easy, moderate, and fast activity compared to the classroom. In other physical settings children were 9.81 (95CI = 3.06, 31.50), 6.91 (95CI = 1.92, 24.95) and 36.90 (95CI = 3.15, >100.00) times as likely to engage in slow easy, moderate, and fast activity compared to the classroom. In the

**Table 4. Research Question 2: Extent of students' participation in Dubai 30x30 activities before, during and after school (data from survey administered to students at schools 2 and 3).**

|     | Physical Education | Classroom | Recess/Other Break Times | Before School | After School | Weekends |
| --- | --- | --- | --- | --- | --- | --- |
| M   | 4.20 | 2.79 | 2.70 | 2.24 | 3.0 | 3.19 |
| SD  | 1.56 | 1.61 | 1.70 | 1.67 | 1.77 | 1.85 |
| N   | 45 | 38 | 40 | 41 | 44 | 44 |

Note: The response scale ranged from 1 ("I did not participate at all in Dubai 30x30 activities") to 6 ("I participated a lot in Dubai 30x30 activities").

multipurpose room children were 4.19 (95CI = 2.05, 8.58) and 4.59 (95CI = 1.81, 11.66) times as likely to engage in limbs and slow easy, but <0.00 (95CI = <0.00, <0.00) times as likely to engage in both moderate and fast activity when compared to the classroom.

Location, instructional setting and activity context were also related to children's activity level. Children were 4.40 (95CI = 2.40, 8.09), 4.13 (95CI = 1.99, 8.57), and 6.55 (95CI = 2.54, 16.92) times as likely to engage in slow easy, moderate, and fast activity when outside or in transition compared to when they were inside. With respect to instructional setting, children were <0.00 (95CI = <0.00, <0.00) times as likely to engage in slow easy, moderate, or fast activity during lunch compared to core class time. Conversely, during music, children were 3.16 (95CI = 1.80, 5.54) and 4.44 (95CI = 1.73, 11.40) times as likely to engage in limbs and slow easy activity, but <0.00 (95CI = <0.00, <0.00) times as likely to engage in moderate or fast activity when compared to core class time. During physical education, children were 3.47

**Table 5. Research Question 3: Relationship between students' PA levels and school context for schools 2 and 3 (multinomial logistic regression of activity levels by OSRAC-E variables).**

|     | n | Limbs | | | Slow Easy | | | Moderate | | | Fast | | |
| --- | --- | --- | --- | --- | --- | --- | --- | --- | --- | --- | --- | --- | --- |
| **Physical Setting** | | | | | | | | | | | | | |
| Classroom (constant) | 848 | - | - | - | - | - | - | - | - | - | - | - | - |
| Gym | 219 | 0.64 | (0.35, | 1.18) | **2.39** | **(1.08,** | **5.28)** | 2.38 | (0.97, | 5.84) | **52.36** | **(13.12,** | **>100.00)** |
| Multipurpose Room | 50 | 4.19 | **(2.05,** | **8.58)** | **4.59** | **(1.81,** | **11.66)** | <0.00 | (<0.00, | <0.00) | <0.00 | (<0.00, | <0.00) |
| Play Group | 40 | 0.99 | (0.22, | 4.54) | **6.02** | **(2.02,** | **17.92)** | 6.48 | (3.49, | 12.02) | **>100.00** | **(31.34,** | **>100.00)** |
| Sports Field | 145 | 1.19 | (0.56, | 2.53) | **5.69** | **(2.65,** | **12.22)** | 4.86 | (1.85, | 12.77) | **55.34** | **(12.52,** | **>100.00)** |
| Other Physical Setting | 67 | 1.06 | (0.59, | 1.89) | **9.81** | **(3.06,** | **31.50)** | 6.91 | (1.92, | 24.95) | **36.90** | **(3.15,** | **>100.00)** |
| **Location** | | | | | | | | | | | | | |
| Inside (constant) | 1,132 | - | - | - | - | - | - | - | - | - | - | - | - |
| Outside or Transition | 237 | **0.92** | **(0.50,** | **1.69)** | **4.40** | **(2.40,** | **8.09)** | **4.13** | **(1.99,** | **8.57)** | **6.55** | **(2.54,** | **16.92)** |
| **Instructional Setting** | | | | | | | | | | | | | |
| Core Class (constant) | 728 | - | - | - | - | - | - | - | - | - | - | - | - |
| Lunch | 49 | 1.09 | (0.23, | 5.19) | <0.00 | (<0.00, | <0.00) | <0.00 | (<0.00, | <0.00) | <0.00 | (<0.00, | <0.00) |
| Music | 40 | **3.16** | **(1.80,** | **5.54)** | **4.44** | **(1.73,** | **11.40)** | <0.00 | (<0.00, | <0.00) | <0.00 | (<0.00, | <0.00) |
| Physical Education | 299 | 0.72 | (0.40, | 1.29) | **3.47** | **(1.76,** | **6.85)** | **3.54** | **(1.53,** | **8.19)** | **>100.00** | **(18.78,** | **>100.00)** |
| Recess | 106 | 1.11 | (0.57, | 2.18) | **3.14** | **(1.21,** | **8.18)** | **3.84** | **(1.52,** | **9.69)** | **>100.00** | **(15.74,** | **>100.00)** |
| Other Instructional Setting | 147 | 1.90 | (0.82, | 4.42) | **3.52** | **(1.10,** | **11.26)** | 3.01 | (0.79, | 11.40) | **46.38** | **(3.93,** | **546.74)** |
| **Activity Context** | | | | | | | | | | | | | |
| Academics | 508 | - | - | - | - | - | - | - | - | - | - | - | - |
| Open Space | 40 | 0.62 | (0.19, | 2.01) | 2.54 | (0.90, | 7.19) | **19.83** | **(4.75,** | **82.79)** | **>100.00** | **(>100.00,** | **>100.00)** |
| Snack | 72 | 1.07 | (0.29, | 3.89) | 0.59 | (0.11, | 3.32) | 5.55 | (0.83, | 36.95) | 1.06 | (0.39, | 2.87) |
| Teacher Arranged | 561 | 1.20 | (0.76, | 1.89) | **5.14** | **(2.64,** | **10.01)** | **15.01** | **(4.06,** | **55.44)** | **>100.00** | **(>100.00,** | **>100.00)** |
| Other Activity Context | 188 | 1.45 | (0.78, | 2.69) | **8.36** | **(3.13,** | **22.33)** | **29.38** | **(6.59,** | **>100.00)** | **>100.00** | **(>100.00,** | **>100.00)** |

(95CI = 1.76, 6.85), 3.54 (95CI = 1.53, 8.19), and >100.00 (95CI = 18.78, >100.00) times as likely to engage in slow easy, moderate, and fast activity compared to core class time. Similarly, during recess children were 3.14 (95CI = 1.21, 8.18), 3.84 (95CI = 1.52, 9.69), and >100.00 (95CI = 15.74, >100.00) times as likely to engage in slow easy, moderate, and fast activity compared to core class time. In other instructional settings, children were 3.52 (95CI = 1.10, 11.26) and 46.38 (95CI = 3.93, >100.00) s likely to engage in slow easy and fast activity compared to core class time.

Regarding activity context, children were 19.83 (95CI = 4.75, 82.79), and >100.00 (95CI = >100.00, >100.00) times as likely to engage in moderate and fast activity during open space activities compared to academic activities. During teacher organized activities, children were 5.14 (95CI = 2.64, 10.01), 15.01 (95CI = 4.06, 55.44), and >100.00 (95CI = >100.00, >100.00) times as likely to engage in slow easy, moderate, and fast activity compared to academic activities. Finally, during other activity contexts children were 8.36 (95CI = 3.13, 22.33), 29.38 (95CI = 6.59, >100.00), and >100.00 (95CI = >100.00, >100.00) times as likely to engage in slow easy, moderate, and fast activity compared to academic activities.

## Research Question 4: Involvement/Engagement of school staff, families and community partners

Table 6 displays the descriptive statistics related to Research Question 4. Overall, physical education teachers' perceived involvement in promoting Dubai 30x30 was higher than classroom teachers' or principals' perceived involvement. Physical education teachers somewhat agreed that they were involved in integrating Dubai 30x30 activities into their lessons, promoting Dubai 30x30 activities outside of physical education, generating support from others to implement Dubai 30x30 activities, finding solutions to problems with implementing Dubai 30x30 activities, planning Dubai 30x30 activities, organizing Dubai 30x30 activities, advocating for the school's involvement in Dubai 30x30 and serving as a health/fitness role model. Principals' perceived involvement was second highest overall in comparison to physical education teachers and classroom teachers. Principals agreed that they were involved with allocating resources for their school's participation in Dubai 30x30 and advocating for their school's involvement in Dubai 30x30. They somewhat agreed that they were involved in evaluating the school's participation in Dubai 30x30, establishing Dubai 30x30 activities at the school, setting performance standards for the school's participation in Dubai 30x30, building/maintaining partnerships for implementing/sustaining Dubai 30x30, planning for Dubai 30x30 implementation and being a healthy/fit role model. Classroom teachers disagreed/somewhat disagreed about their involvement across all relevant items.

Themes and supporting evidence related to Research Question 4 are presented in Table 7. The first theme, "Physical Education Teachers as Dubai 30x30 Leaders," closely aligns with the survey data, further demonstrating that physical education teachers were distinctly involved in schools' implementation of the initiative. The common approach was for someone on the physical education team to spearhead the planning and organization of Dubai 30x30 activities, guide the other physical education teachers in their efforts to support the implementation of the activities and serve as the primary liaison for other school staff, parents and any other entities who were involved. The second them focused on parent engagement, which mainly entailed parents participating in various activities that the school provided, some of which were specifically designed for parents.

**Table 6. Research Question 4: Involvement/Engagement of school staff in promoting Dubai 30x30 (data from surveys administered to physical education teachers, classroom teachers and principals at schools 2 and 3).**

| Survey Item | Physical Education Teachers | | | Classroom Teachers | | | Principals | | |
|---|---|---|---|---|---|---|---|---|---|
| | M | SD | N | M | SD | N | M | SD | N |
| I am involved with integrating Dubai 30x30 activities into physical education lessons (PE only) | 4.72 | 1.53 | 18 | N/A | N/A | N/A | N/A | N/A | N/A |
| I am involved with promoting Dubai 30x30 activities outside of physical education (PE only) | 4.50 | 1.50 | 18 | N/A | N/A | N/A | N/A | N/A | N/A |
| I am involved with generating support from others (e.g., parents, other teachers) to implement Dubai 30x30 activities (PE and Classroom only) | 4.55 | 1.46 | 18 | 2.95 | 1.28 | 20 | N/A | N/A | N/A |
| I am involved with securing resources to implement Dubai 30x30 activities (PE and Classroom only) | 3.94 | 1.52 | 17 | 2.16 | 1.21 | 20 | N/A | N/A | N/A |
| I am involved with collaboration among others (e.g., colleagues, parents, community partners) to maximize my school's provision of Dubai 30x30 activities (PE and Classroom only) | 4.44 | 1.65 | 18 | 2.60 | 1.35 | 20 | N/A | N/A | N/A |
| I am involved with finding solutions to problems that could limit my school's ability to provide Dubai 30x30 activities (PE and Classroom only) | 4.33 | 1.41 | 18 | 2.11 | 1.56 | 20 | N/A | N/A | N/A |
| I am involved with integrating Dubai 30x30 activities into my classroom lessons or as breaks/transitions during normal classroom time (Classroom only) | N/A | N/A | N/A | 3.10 | 1.41 | 20 | N/A | N/A | N/A |
| I am involved with providing Dubai 30x30 activities during recess or other break times during school (Classroom only) | N/A | N/A | N/A | 3.05 | 1.32 | 20 | N/A | N/A | N/A |
| I am involved with evaluating my school's participation in Dubai 30x30 (Principals only) | N/A | N/A | N/A | N/A | N/A | N/A | 4.50 | 0.71 | 2 |
| I am involved with providing professional development opportunities related to Dubai 30x30 at my school (Principals only) | N/A | N/A | N/A | N/A | N/A | N/A | 3.00 | 2.83 | 2 |
| I am involved with establishing Dubai 30x30 activities at my school (Principals only) | N/A | N/A | N/A | N/A | N/A | N/A | 4.00 | 0 | 2 |
| I am involved with setting performance standards for my school's participation in Dubai 30x30 (Principals only) | N/A | N/A | N/A | N/A | N/A | N/A | 4.00 | 0 | 2 |
| I am involved with allocating resources for my school's participation in Dubai 30x30 (Principals only) | N/A | N/A | N/A | N/A | N/A | N/A | 5.00 | 0 | 2 |
| I am involved with building/maintaining partnerships with community constituents to implement/sustain my school's participation in Dubai 30x30 (Principals only) | N/A | N/A | N/A | N/A | N/A | N/A | 4.00 | 0 | 2 |
| I am involved with Dubai 30x30 planning at my school | 4.72 | 1.23 | 18 | 2.25 | 1.45 | 20 | 4.50 | 0.71 | 2 |
| I am involved with serving on my school's Dubai 30x30 committee or other related (e.g., school wellness) committee, board, or task force | 3.11 | 1.91 | 18 | 2.11 | 1.56 | 20 | 3.00 | 1.41 | 2 |
| I am involved with organizing Dubai 30x30 activities at my school | 4.39 | 1.33 | 18 | 2.35 | 1.37 | 20 | 2.00 | 0 | 2 |
| I am involved with staying up to date on best practices for school health/wellness programming in relation to my school's participation in Dubai 30x30 | 3.78 | 1.66 | 18 | 2.85 | 1.42 | 20 | 3.50 | 0.71 | 2 |
| I am involved with advocating for my school's participation in Dubai 30x30 | 4.28 | 1.49 | 18 | 3.63 | 1.57 | 20 | 5.50 | 0.71 | 2 |
| I am involved with being a health/fitness role model for others in my school during Dubai 30x30 | 4.83 | 1.20 | 18 | 3.25 | 1.68 | 20 | 4.50 | 0.71 | 2 |
| *Total Involvement Score* | 4.31 | N/A | N/A | 2.73 | N/A | N/A | 3.96 | N/A | N/A |

Note: The response scale ranged from 1 ("Strongly Disagree") to 6 ("Strongly Agree").

## Research Question 5: School staff's perceived successes and challenges of Dubai 30x30 implementation

Themes and supporting evidence related to Research Question 5 are presented in Table 8. Implementation successes were described in terms of communities coming together. Implementation challenges were mainly discussed relative to the lack of guidance and the pressures placed on physical education staff.

**Table 7. Research Question 4: Involvement/Engagement of school staff, families and community partners in promoting Dubai 30x30 (themes and supporting evidence from school staff focus group and interview data from all three schools).**

| Theme | Supporting Evidence |
|---|---|
| Physical Education Teachers as Dubai 30x30 Leaders | "So, I was the Dubai30/30 champion as that, that's what they called the lead role in the school. So, I was the one that coordinated everything in the school basically and just I had my, you know the PE Team. I, I didn't obviously tell them what to do, but anything I needed support with, or I needed them to organize or help with. I would, you know, give them the instructions there, the emails to parents would come from me any communication I would be passing on. So yeah, just to coordinate you just the coordinator of it for your whole school." (PE Teacher 1, School 3) <br> "[PE Teacher 1, School 3] drove the whole thing. . .Gave us a set of some guidelines and I'd say most of us chipped in with different ideas. All of us definitely supported the events, things that happened whether it was accompanying the kids on the that trip, whether it was getting involved. . .in fact all of us helped with the night run, which is a really, from a parental perspective, really launched the whole connection between people–staff and parents–with the 30/30 (PE Teacher 2, School 3) |
| Parent Engagement | "We did a parent challenge where we ran fitness [activities] at drop off, and children nominated their parents to be part of that. . . parents [came] in and participat[ed] in our morning runs, and our morning cycles, [and] in the night run." (Principal, School 3) <br> "The Wake Up, Shake Ups–all the parents coming in and getting involved was lovely, lovely to see." (PE Teacher 1, School 1) <br> "A lot of staff got a little bit more motivated to try to get involved with other [activities] because they saw the positive response from the kids and the parents" (PE Teacher 3, School 3) <br> "My mother and dad are really into Dubai 30x30, and they are also involved" (Student 1, School 2) |

**Table 8. Research Question 5: School staff's perceived implementation successes and challenges (themes and supporting evidence from school staff focus group and interview data from all three schools).**

| Theme | Supporting Evidence |
|---|---|
| Communities Coming Together | "It was a really good focus to bring the community together" (Principal, School 3) <br> "It's a good sense of community, really brings everybody together. And there's such a wide range of activities available, even if you're not the most sporty person, it's something that everybody can enjoy" (Classroom Teacher 3, School 3) <br> "It felt like we were all kind of one big team. . .all of the staff, the parents and the pupils all participating together. It was nice that we were all working together. Everyone had the same goal of getting the 30 minutes in" (PE Teacher 2, School 3) <br> "I think it makes people come together as a community. . .it opens up the field for a lot of activities that involve children, parents and staff" (Classroom Teacher 2, School 3) |
| Lack of Guidance | "There aren't any set guidelines as to how much a school needs to be doing. . .there needs to be a little bit more clarity from [the government] as to what the targets for each school [should be] because I know that we did a lot, but then I spoke to another school and they literally just had a poster" (PE Teacher 3, School 3) <br> "I took on [being the Dubai 30x30 champion for the school], not really knowing (being my first year) what I was meant to be doing" (PE Teacher 1, School 1) |
| Pressures on Physical Education Staff | "It is a big task for just one department to do. . .It's a lot lumped into one month. . .you feel, you know, you've got to keep banging that drum in between doing everything you do in your day job anyway" (PE Teacher 3, School 3) <br> "The main challenge for us was just the timing of our daily schedules and the [school inspections by the government] and then trying to do this additionally. . .I did feel I wanted to do more, and I physically couldn't because I had not time" (PE Teacher 1, School 1) <br> "I don't think we found any challenges [although] it's just fitting everything into a busy time, isn't it? That's the thing" (Principal, School 3) |

### Research Question 6: Participants' perceived impact of Dubai 30x30

The themes and supporting evidence related to Research Question 6 are displayed in Table 9. There were two themes: (a) increased PA and (b) promotion of the school. Many of the statements participants made about the impact of Dubai 30x30 focused on increased PA for youth and parents. Promotion of the school constituted a smaller theme that became apparent within the transcripts from participants at School 3.

## Discussion

Consistent with global trends in PA among school-age youth [3], most children and adolescents in the UAE are insufficiently active [23]. The current study drew upon conceptions of whole-of-school PA promotion [7–9] to examine schools' participation in Dubai 30x30. To our knowledge, this is the first study to provide descriptive data about implementation and outcomes related to schools' involvement with Dubai 30x30. We therefore adopted an exploratory approach, driven by several research questions intended to provide an initial glimpse into various aspects of whole-of-school PA. Specifically, this study investigated the scope and nature of Dubai 30x30 activities offered before, during and after school and on weekends; students' participation in Dubai 30x30 activities; the role of school contextual variables in students' PA during Dubai 30x30; staff involvement in promoting Dubai 30x30; staff's perceptions of implementation successes and challenges; and staff's and students' perceptions of the impact of Dubai 30x30. The results of this study contribute to the growing knowledge base internationally on whole-of-school PA approaches, point to potential directions for future research, and establish preliminary evidence that can be used to optimize Dubai 30x30 as a lever for increasing and sustaining whole-of-school PA in Dubai British curriculum schools.

**Table 9. Research Question 6: Perceived impact of Dubai 30x30 (themes and supporting evidence from focus group and interview data from all three schools).**

| Theme | Supporting Evidence |
|---|---|
| Increased PA | "I think Dubai 30x30 started my consistency [with exercise] because before, I didn't really exercise but now I'm more consistent with it" (Student 1, School 2) <br> "I actually started wanting to work out more. I started going to the gym…now I go two times a week. I know it's definitely affected my brother a lot. He also does gym now. He started like this whole plan and everything. I feel like the things you do and if you find them enjoyable at school you are most likely to do them outside of school" (Student 2, School 2) <br> "There were children that would come to me and go, 'Miss, I did this at the weekend' or 'I did that at the weekend' and you know that Dubai 30x30's inspired them to do that and that wouldn't have happened maybe before" (PE Teacher, School 1) <br> "Wake Up, Shake Ups" we still do now. So there's been a lasting impact" (Classroom Teacher 2, School 3) <br> "It's getting back into exercise. I think it's the timing of the year but also, we had a bike and treadmill at reception and there were a lot of parents that came in and said, 'I haven't cycled for years, and I think the impact, you know, we've got huge cycle tracks [in Dubai] and [parents] went out and continued. The legacy of the Dubai 30x30 continues afterwards. [People] got back into exercise afterwards" (Classroom Teacher 4, School 3) |
| Promotion of the School | "…the assemblies and the guests coming in and the stair climbs…it's just something different. Yeah, and it's a good way to promote the school" (PE Teacher 3, School 3) <br> "Across the whole school, just the optics were very good. Everywhere you looked, you could see posters on the walls. When you walked into reception, there is the cycle bike and there is the treadmill and everything. So, it's everywhere you went you would just know. If someone came on holiday and they came to visit the school on their first day in Dubai, they'd probably ask 'What is this 30/30?'" (PE Teacher 2, School 3) <br> "We have more parental involvement, we have more activities for the children on a daily basis…running through the school with posters on our interactive screens" (Principal, School 3) |

### Research Question 1: Dubai 30x30 activities provided

School staff reported that most Dubai 30x30 activities were provided in physical education, during school break times and before and after school. This suggests that Dubai 30x30 activities spanned many of the contexts identified in whole-of-school PA recommendations [7–9]. Internationally, there are inconsistencies in the degree to which schools provide PA through all possible recommended components/practices of a whole-of-school approach [10,11,14,15,38–40]. For example, in a national study of U.S. secondary schools, 7% of middle school students and less than 1% of high school students reported attending schools that offered all six practices conceptualized as a whole-of-school approach (having physical education; offering PA breaks during the school day; having intramural sports; having interscholastic sports; having active transportation opportunities to/from school; and having shared-use of school facilities agreements with outside entities; [11]. The results of a meta-analysis study indicated that increasing the number of components through which school-based PA programming is provided is positively associated with an increase in students' total daily PA minutes [41]. However, Webster et al. [42] assert that school-based PA programming should be viewed as sufficient based on whether all students achieve the goals of the program, as opposed to whether the program consists of a certain number of components or practices. Future research on schools' implementation of Dubai 30x30 should further investigate both the quantity and quality of PA offerings and the association of such offerings with targeted student outcomes (e.g., meeting PA guidelines, meeting academic standards).

### Research Question 2: Students' participation in Dubai 30x30 activities

Students reported that they mainly participated in Dubai 30x30 activities during physical education and occasionally participated in activities after school and on weekends but infrequently participated in activities offered during break times. Comparing these data to the results for the first research question, it seems that providing opportunities for PA during break times may not always lead to students participating in such opportunities. This might have to do with students' motivation. Lonsdale et al. found that adolescents in Hong Kong who were more self-determined in their motivation to participate in physical education had higher PA levels during a free-choice period compared to students who were less self-determined in their motivation [43]. Therefore, a possible direction for future research on schools' participation in Dubai 30x30 is to examine the relationship between students' motivation and their participation in Dubai 30x30 activities. Additionally, the qualities or characteristics of Dubai 30x30 activities that students find most appealing, and which serve as sources of motivation to participate warrant increased attention in future studies. The present study revealed that students were drawn to competitive activities. Other whole-of-school research also highlighted competition as a key driver for students' involvement in PA opportunities, although recognition of participation and effort was also important for fostering involvement [21]. In another study, some adolescents' PA levels were highest when non-competitive physical activities were provided as part of a comprehensive approach to school health [44].

### Research Question 3: Relationship of students' PA to school contextual variables

During school, students were more likely to reach higher PA intensity levels when they were in locations, instructional settings, and activity contexts other than the regular classroom setting, core class time and academic activities. These findings reinforce the data addressing the first two research questions in that the participating schools appeared to place a limited focus on

Dubai 30x30 as a catalyst or incubator for increasing PA during regular classroom time. Providing PA opportunities during periods of the day devoted to academic instruction (outside of physical education)–a practice referred to as movement integration [45]–is recommended as part of a whole-of-school PA approach [7–9]. The current literature on movement integration mostly centers on primary/elementary school settings. In such contexts, movement integration in can involve various strategies, such as infusing PA into academic lessons or giving students brief breaks from sitting [46]. Previous research shows that integrating PA during regular classroom time increases children's PA participation and educational outcomes [47]. However, classroom teachers often report barriers to using movement integration (e.g., lack of time, lack of resources, lack of support from school administrators) [48,49]. It would be interesting in future research to further explore the potential of using regular classroom time during Dubai 30x30 to increase children and adolescents' school-based PA.

## Research Question 4: Involvement/Engagement of staff, families and community partners in Dubai 30x30

Among school staff, physical education teachers were most involved and classroom teachers were least involved in promoting Dubai 30x30, which resonates with the results for the first three research questions. Whole-of-school PA recommendations commonly call upon physical education teachers to serve as leaders in providing PA opportunities and galvanizing the support of other school staff, parents, and community partners [50,51]. Yet, Webster et al. note that in many cases, certain obstacles may need to be overcome for physical education teachers to effectively lead whole-of-school PA initiatives [52]. Such obstacles include a lack of professional preparation and training specific to whole-of-school PA leadership and a lack of incentives for physical education teachers to take on leadership responsibilities (e.g., personal interest, tangible rewards, external accountability). The findings in the present study seem to suggest that physical education teachers were successful in garnering involvement from parents and engaging numerous community partners. Principals also provided support for schools' participation in Dubai 30x30. Thus, a priority for future implementations of the initiative will be to increase attention to helping physical education teachers build enhanced support from classroom teachers.

## Research Question 5: Staff's perceived implementation successes and challenges

Staff perceived that Dubai 30x30 brought their school communities together. This is a fundamental goal of whole-of-school PA approaches, which are intended to draw upon the collaboration and synergy of school staff, families, and communities [7–9] However, physical education teachers also commented on the lack of guidance concerning implementation and they felt burdened by the amount of work involved and pressure to make Dubai 30x30 a special month for their schools. Unfortunately, we neglected to more fully explore the source of the pressure physical education teachers felt in relation to implementing Dubai 30x30 activities. In previous research, factors at multiple social-ecological levels exerted an influence (e.g., public policy, administrative support, teacher beliefs) on whole-of-school PA practices [19,20]. In the case of Dubai 30x30, which the Dubai government initiated and strongly promotes, it is possible that the pressure physical education teachers feel may stem from the government, but it is also possible that other factors, such as the overall school ethos, principals' priorities, parents' interest, or physical education teachers' own personal values contribute to these feelings of pressure.

Although principals noted that time was a barrier to implementation, neither they nor the classroom teachers communicated feelings of being overwhelmed, in contrast to what was evidenced in the focus groups and interview with physical education teachers. A more balanced diffusion of responsibility across all school stakeholders should be sought in future school-based implementations of the Dubai 30x30. For instance, consistent with recommendations for implementing whole-of-school PA programming [27], schools could form a school wide Dubai 30x30 committee to promote distributed leadership, shared decision-making, and clear guidance with respect to the initiative. Such a committee could a designated Dubai 30x30 "champion" (e.g., a physical education teacher), classroom teacher representatives, a school administrator, parent representatives, student representatives and community partner representatives.

## Research Question 6: Perceived impact of schools' implementation of Dubai 30x30

Participants believed Dubai 30x30 increased PA participation and helped to promote their schools. The value of these beliefs in energizing school staff and sustaining their involvement in the initiative is an important area of focus for future research. A study of school principals in the U.S. found that participants' outcome expectations directly predicted their level of self-reported involvement in CSPAPs [30]. Specifically, principals were more likely to be involved if they believed a CSPAP would benefit students not just in terms of their PA participation but also in terms of their physical development, cognitive development, social development, school attendance and academic performance. Participants in the present study made little reference to most of these outcomes when discussing the impact of Dubai 30x30. In the future, researchers should more systematically measure a broad range of student outcomes associated with schools' participation in Dubai 30x30, as such data may spur staff's involvement.

## Limitations

This study had several limitations. First, although working with only a few schools allowed us to explore school-based Dubai 30x30 implementation from multiple perspectives, the use of a small convenience sample limits the generalizability of the results. Further research is needed to document Dubai 30x30 implementation in a broader range of both private and public schools in Dubai. Second, we sought to survey parents in this study, but no parents participated. Understanding the perspectives of parents and community partners is critical to advancing research and practice related to schools' implementation of Dubai 30x30 and to the implementation of other initiatives that lend insights into whole-of-school PA programming. Third, since the individuals who collected observation data at the schools were student volunteers, their academic schedules sometimes prevented them from visiting the schools at the desired times. Through funded research, it may be possible to adhere to a more regular and extensive observation schedule. Fourth, this study merely provides a snapshot of schools' participation in the initiative. Future studies should employ multi-year data collection protocols to examine participation patterns and investigate factors that may be associated with variation in implementation rates and strategies. Furthermore, studies adopting experimental designs will be useful in determining cause-effect relationships between implementation practices and outcomes.

## Conclusions

This study provides an initial glimpse into British curriculum schools' implementation of Dubai 30x30 from a whole-of-school PA perspective. In line with recommended whole-of-

school PA practices [7–9], physical education served as a central component in the promotion of PA within the participating schools and physical education teachers played leading roles in organizing and implementing Dubai 30x30 activities. While implementation resulted in several positive outcomes, physical education teachers bore the brunt of responsibility for schools' participation in the initiative and a gap in PA promotion was evident during regular classroom time. Further research is needed to determine whether these results generalize to other schools in Dubai, and if so, how to optimally support collaborations between physical education teachers and other school stakeholders to harness Dubai 30x30 for whole-of-school PA. Overall, this study underscores the utility of using a whole-of-school PA lens to frame schools' implementation of Dubai 30x30 and gain key insights about processes and outcomes that warrant particular attention for strengthening school communities' engagement in Dubai 30x30 and other initiatives that focus on increasing the PA of children and adolescents.

## Supporting information

**S1 File. Inclusivity in global research questionnaire.**
(PDF)

**S2 File. Survey results for all teachers and principals.**
(XLSX)

**S3 File. Student survey results (post).**
(XLSX)

**S4 File. Observation data.**
(XLSX)

## Acknowledgments

The authors wish to acknowledge the support of the principals, teachers and students who agreed to participate in this study.

## Author Contributions

**Conceptualization:** Chris McMahon, Collin A. Webster.

**Data curation:** Chris McMahon, Collin A. Webster.

**Formal analysis:** Chris McMahon, Collin A. Webster, R. Glenn Weaver.

**Investigation:** Collin A. Webster, Christophe El Haber, Gönül Tekkurşun Demir, Zainab Mohamed Ismail, Syeda Zoha Fatima Naqvi, Mehnaz Ghani, Şevval Kepenek, Manel Kherraf, Thrisha Krishnakumar, Pranati Prakash, Yeowon Seo.

**Methodology:** Collin A. Webster.

**Supervision:** Collin A. Webster.

**Writing – original draft:** Chris McMahon, Collin A. Webster, R. Glenn Weaver.

**Writing – review & editing:** Chris McMahon, Collin A. Webster, R. Glenn Weaver, Christophe El Haber, Gönül Tekkurşun Demir, Zainab Mohamed Ismail, Syeda Zoha Fatima Naqvi, Mehnaz Ghani, Şevval Kepenek, Manel Kherraf, Thrisha Krishnakumar, Pranati Prakash, Yeowon Seo.

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
