## [Decision Letter · Decision Letter 0]

29 Nov 2023

PONE-D-23-25904Whole-of-School Physical Activity Implementation in the Context of the Dubai Fitness ChallengePLOS ONE

Dear Dr. Webster,

Thank you for submitting your manuscript to PLOS ONE. After careful consideration, we feel that it has merit but does not fully meet PLOS ONE’s publication criteria as it currently stands. Therefore, we invite you to submit a revised version of the manuscript that addresses the points raised during the review process.

We look forward to receiving your revised manuscript.

Kind regards,

Josie N. Booth

Academic Editor

PLOS ONE

Journal Requirements:

Reviewers' comments:

Reviewer's Responses to Questions

**Comments to the Author**

1. Is the manuscript technically sound, and do the data support the conclusions?

Reviewer #1: Yes

Reviewer #2: Partly

2. Has the statistical analysis been performed appropriately and rigorously? 

Reviewer #1: Yes

Reviewer #2: Yes

3. Have the authors made all data underlying the findings in their manuscript fully available?

Reviewer #1: No

Reviewer #2: No

4. Is the manuscript presented in an intelligible fashion and written in standard English?

Reviewer #1: Yes

Reviewer #2: Yes

5. Review Comments to the Author

Reviewer #1: The manuscript has been written in a style that is readable and easy to understand. There are several strong points as highlighted in my review report. However, there are some weaknesses that have been outlined that authors need to address. They include ethical issues. Details are in the attached report. The authors have also indicated that some restrictions will apply on the availability of data.

Reviewer #2: Generally, it is an interesting manuscript. It has the potential to add the body of knowledge in the health promotion field. With regards to the research idea, the authors have to be applauded for their interest in exploring how Dubai 30x30 initiative is seen from the whole-of-school approach. However, there are some major (mostly on the study design) and minor points need to be addressed:

1. While the purpose of the study is to examine schools’ participation in an annual, government-led, and emirate-wide initiative in Dubai, it was not clear if it was intended as evaluation research. The authors employed a mixed-method including surveys, observation and Focus Group Discussion, but the focus was simply to the extent to which students, teachers, parents and schools have participated in the initiative. However, the participation itself was not clearly defined, e.g., whether the students engaged in physical activity as per their usual (PE class) or because they intended to participate in the Dubai 30x30. There was also no comparison - before and after exposure (Dubai 30x30) assessment to justify that there was a behavioral change observed after the initiative.

2. L192: The design of the study, particularly on sampling methods, was rather vague. In the introduction part, the authors mentioned the participation rate of Dubai 30x30 was high, yet convenience sampling was used in the study. The reasons of involving only three private international schools were also absent. Why only three, and why only international schools? The authors should also mention how difference the characteristics of these three schools are compared to Dubai schools in general. The largest nationality group of students at School 1 was UK, while the largest nationality group of students for School 2 was European. What is the implication of this study to Dubai nationals in general?

3. Line 210-Table 1: Participant selection was also unclear. While the Dubai 30x30 initiative itself has gained a lot of attention and high participation was reported, this study only involved 45 students and 40 school principals and teachers in the survey. The authors should clarify the selection criteria/recruitment of the schools, students and staff.

4. Table 1 and L231: It was mentioned in the beginning of the methods section that three schools were involved in the study, but only 2 schools were involved in the survey and observation. Please clarify which schools were involved in each method of data collection. Adding one column in table1 would also be better to provide the readers clarification of the participants' affiliation. For example, there were 45 students involved in the survey, but it was unclear which schools they were from.

5. L299-302: The authors mentioned they scheduled the observation in accordance with when the observers were available and when the schools were able to accommodate observers. This may lead to bias in the timing. How do the authors determine the observation period? Whether it’s only observed during the day when PE was taught or randomly/systematically selected during the week? Was the observation period uniform between different schools? Observing a class in school A in the morning on Tuesday PE class (9-11am) would result in significantly different behavior of the students in school B who were observed on Wednesday afternoon 2-4pm (their normal class). Some clarifications on this matter should be added.

6. L487-488: Please note that the two schools involved are international schools with British or European children as majority. In the discussion, the authors should also address the limitation in the generalization of this study considering the convenience sampling being used and the selection of the participants.

7. L514-518: the authors should further discuss to what extent the activities are different from their regular practice. The authors should also mention and discuss, what kind of activities were provided in the PE class, during breaks and before/after school. Were the activities also provided before Dubai 30x30 initiatives? If yes, does it mean schools were just do as their normal? Or is there any new activities that schools have done to participate in the Dubai 30x30 initiatives? (crosscheck the arguments with the students’ responses on motivation).

8. L381: The findings showed an inconsistent result, between teachers’ vs students’ perspective regarding their participation. This should be discussed further; whether the teachers responded normatively as they are expected to show school participation?

Few minor points to help improve the readers’ understanding:

9. Tables should be self-explanatory, but I find it difficult to understand the tables presented. What is the unit of M in table 2? Please clarify whether it’s number of activities, or minutes? Was it times per week? Or per month? Please also clarify how many schools are involved.

10. Please also clarify the unit of table 4. See previous comment for table 2: 45 students from how many schools?

11. Table 5: Relationship between Students’ PA Levels and School Context. The outcome of the study was not clear; why physical activity of the students was assessed on its association with the school contexts, but not on how it’s related to Dubai 30x30?

12. State the unit of M in Table 6. What does the score mean?

6. PLOS authors have the option to publish the peer review history of their article (what does this mean?). If published, this will include your full peer review and any attached files.

Reviewer #1: No

Reviewer #2: No

---

## [Author Response · Author response to Decision Letter 0]

18 Dec 2023

Authors’ Responses to Reviewers’ Comments

Dear Dr. Booth,

Thank you for the opportunity to address the reviewers’ comments and revise our manuscript. We appreciate the careful attention the reviewers gave to our work, and we feel that our revisions have considerably strengthened the manuscript as a result. Our responses to reviewer comments are uploaded as a separate file. Responses are in bold and highlighted in yellow. Our responses can also be found below. Our revisions in the uploaded manuscript are highlighted in yellow.

Thank you for your continued consideration of this study for publication in PLOS One.

Sincerely,

Collin A. Webster, Ph.D.

Texas A&M University – Corpus Christi

colllin.webster@tamucc.edu

+1 (361) 443-4789

Reviewer #1: The manuscript has been written in a style that is readable and easy to understand. There are several strong points as highlighted in my review report. However, there are some weaknesses that have been outlined that authors need to address. They include ethical issues. Details are in the attached report. The authors have also indicated that some restrictions will apply on the availability of data.

Author Response: Thank you for your encouraging remarks and careful review of the manuscript. We hope our responses and revisions have adequately addressed each comment.

1. Introduction

The report is composed of different aspects that are necessary for decision-making regarding the readiness of the manuscript to be considered for publication by the PLOS ONE Journal. It outlines the introduction, summary of the research and my overall impression, the strong points, discussion of specific areas for improvement (weaknesses, major issues and minor issues), other points that need to be considered, and recommendations.

2. Summary of the research and my overall impression

Even though the manuscript has some weaknesses that will be outlined later, it has strong points. Consequently, my overall impression of the manuscript is that it is publishable in the PLOS ONE Journal. The manuscript is based on the study that was conducted focusing on the Whole-of- school Physical activity in Dubai.

3. Strong points of the manuscript

The strong points are that:

1. The manuscript is in line with the aim, the scope and publication criteria of the Journal. The number of words on the abstract is within the acceptable range. It focuses on an area that is very important for different stakeholders, which include students, teachers and the community regarding physical activity. 

2. There is consistency in terms of the way in which sources have been referenced.

3. Good attempt in terms of the way in which the manuscript has been edited.

4. The knowledge gap has been clearly identified in page 7, line 171 and line 172.

5. Ethical issues on informed consent and consent from parents/guardians have been included as outlined in line 224 and line 225.

6. There is a clear indication of the way in which each of the research questions have been answered.

7. There is a detailed explanation of the way in which data were generated and analysed to answer the research questions.

8. Quotations have been used to give voice to the participants

9. Sources have been used under discussion to either support or negate the argument

10. The authors have outlined the limitations of the study. 

11. The conclusion is based on the data, results and the discussion section of the study. However, it can be improved by including other aspects in line with the research questions and themes that were generated. The proposal for the improvement has been outlined under weaknesses.

 4. Discussion of specific areas for improvement (weaknesses, major issues and minor issues)

Weaknesses 

Strong points notwithstanding, the manuscript has weaknesses that need to be addressed to improve its quality and acceptability for publication by the PLOS ONE Journal. Weaknesses are that:

4.1 Major Issues

1. The study used the Whole-of-School Physical Activity approach as a lens; however, there is no indication of the way in which the approach guided the authors with regard to the selection of the research design, sampling and research methods. 

Author Response: We have added details in the manuscript to more clearly tie our use of a whole-of-school lens to our research questions and methodological approach.

2. The authors have selected a convergent parallel mixed-methods research design; however, there is no justification for the selection of this kind of a design.

Author Response: We have provided an explanation for choosing this research design. 

3. In page 10, line 222, the authors have blocked the name of the ethics board that gave ethical clearance for the study, and that breaks the flow of the sentence. I suggest that the authors use a pseudonym to represent the name of the institution than to block it. 

Author Response: We have removed the block and used a pseudonym for our institutional ethics board.

4. In line 236 and 237, the authors indicate that the students were able to complete the survey during normal classroom time with the assistance of teachers. The question is “Did the assistance by teachers not compromise the data?” Which steps were taken to ensure that data generation is not compromised? Did the assistance by teachers during normal teaching time not disrupt teaching and learning by teachers and students?

Author Response: We have added details here to address these questions.

5. What were the disadvantages of using volunteer participants as reflected in page 13, line 279?

Author Response: These individuals were not participants in the study. They were undergraduate students who volunteered to assist with data collection and other aspects of the study as part of the research team. They did not receive any extra credit for their assistance. They felt the opportunity to help with the study was important to their career goals, as they had plans to pursue advanced degrees and potentially lead research projects in the future. Therefore, these individuals facilitated this research in numerous ways. The only disadvantage was that the volunteers’ schedules sometimes conflicted with times during which we wanted to conduct observations at the schools. We have added this limitation to our discussion section, although we wish to note that our aim in the study was to assemble an overview of the participating schools’ implementation of Dubai 30x30, combining and triangulating evidence from multiple data sources, rather than to comparing/contrasting each school’s experience.

4.2 Minor issues

There may be a need to include parental engagement and the usefulness of using a whole-of-school PA lens on a study in the conclusion section.

Author Response: We mentioned in our limitations that the lack of parent participants limits the comprehensiveness of the study and that future investigations would benefit from obtaining parents’ perspectives of Dubai 30x30 and other initiatives that can inform research and practice related to whole-of-school PA. Since our study did not include data from parents, and the manuscript is already quite long, we did not feel further discussion of parent engagement is warranted. However, we did add some perspective in the conclusion that underscores the importance of using a whole-of-school PA lens to better understand schools’ implementation of Dubai 30x30 and other initiatives where a systems approach is used to increase physical activity engagement.

5. Other points

The authors have also omitted other ethical aspects that are necessary for the study of this nature. For example, privacy and confidentiality, anonymity, benefits, risks and harm, and assent from the students.

Author Response: We have added this information to the methods section.

6. Recommendations

My recommendation is minor revisions 

Reviewer #2: Generally, it is an interesting manuscript. It has the potential to add the body of knowledge in the health promotion field. With regards to the research idea, the authors have to be applauded for their interest in exploring how Dubai 30x30 initiative is seen from the whole-of-school approach. However, there are some major (mostly on the study design) and minor points need to be addressed:

Author Response: We wish to thank the reviewer for these encouraging remarks and hope that our responses and revisions adequately address the comments below.

1. While the purpose of the study is to examine schools’ participation in an annual, government-led, and emirate-wide initiative in Dubai, it was not clear if it was intended as evaluation research. The authors employed a mixed-method including surveys, observation and Focus Group Discussion, but the focus was simply to the extent to which students, teachers, parents and schools have participated in the initiative. However, the participation itself was not clearly defined, e.g., whether the students engaged in physical activity as per their usual (PE class) or because they intended to participate in the Dubai 30x30. There was also no comparison - before and after exposure (Dubai 30x30) assessment to justify that there was a behavioral change observed after the initiative.

Author Response: Thank you for pointing out this important distinction. We have now clarified in our methods section (study design) that the study is descriptive in nature and that we did not intend to conduct an evaluation or to determine changes across time (e.g., via an experimental design). Our aim was to document processes and outcomes that characterized the implementation of Dubai 30x30 in 2022. Based on our data collection methods (e.g., interview and focus group questions), our data likely reflect participants’ involvement encompassed activities that were mainly unique to Dubai 30x30.

2. L192: The design of the study, particularly on sampling methods, was rather vague. In the introduction part, the authors mentioned the participation rate of Dubai 30x30 was high, yet convenience sampling was used in the study. The reasons of involving only three private international schools were also absent. Why only three, and why only international schools? The authors should also mention how difference the characteristics of these three schools are compared to Dubai schools in general. The largest nationality group of students at School 1 was UK, while the largest nationality group of students for School 2 was European. What is the implication of this study to Dubai nationals in general?

Author Response: Thank you for carefully considering our approach to sampling for the study. We have added information about sampling in our methods section. Specifically, we have added that although participation rates in Dubai 30x30 are reported to be high, access to schools for research can prove challenging. We drew upon our existing partnerships with British curriculum schools to recruit schools to participate in our research. We initially aimed to include more than three schools in our study, but several schools we invited to participate declined. As we state in our discussion, we acknowledge that there are limitations with our sample, but we also feel that our use of multiple quantitative and qualitative methods has allowed for an in-depth investigation of our participant schools’ implementation of Dubai 30x30. 

Regarding the reviewer’s comment about how our participant schools compare to other schools in Dubai, we have provided further contextual details in our methods section. 

Given the focus of our study on only three schools, all of which are private/international British curriculum schools, we feel further research on a broader scale is needed to develop implications for Dubai nationals in general.

3. Line 210-Table 1: Participant selection was also unclear. While the Dubai 30x30 initiative itself has gained a lot of attention and high participation was reported, this study only involved 45 students and 40 school principals and teachers in the survey. The authors should clarify the selection criteria/recruitment of the schools, students and staff.

Author Response: Please refer to our response to the reviewer’s previous comment concerning our sampling strategy for school recruitment. Regarding the selection of participants within each school, we have added further details about this to our methods section. Specifically, we added that all principals, teachers, and students were eligible to participate in study and were invited to do so; however, as participation was voluntary and there were no tangible participation incentives, our total number of participants was smaller than we hoped for. Even so, we feel that our multiple quantitative and qualitative data collection techniques yielded a robust data set for the study.

4. Table 1 and L231: It was mentioned in the beginning of the methods section that three schools were involved in the study, but only 2 schools were involved in the survey and observation. Please clarify which schools were involved in each method of data collection. Adding one column in table1 would also be better to provide the readers clarification of the participants' affiliation. For example, there were 45 students involved in the survey, but it was unclear which schools they were from.

Author Response: We have added information to detail which schools were involved with which data collection methods. Unfortunately, we did not ask students to report which school they attended. While this information would yield interesting insights, we wish to emphasize that our aim in this study was to assemble a comprehensive view of Dubai 30x30 implementation and outcomes at the participating schools using multiple quantitative and qualitative data collection methods. Rather than investigate the different schools as individual case studies, we were interested in identifying commonalities and consistencies to glean an overall picture of the implementation experience across the participating schools.

5. L299-302: The authors mentioned they scheduled the observation in accordance with when the observers were available and when the schools were able to accommodate observers. This may lead to bias in the timing. How do the authors determine the observation period? Whether it’s only observed during the day when PE was taught or randomly/systematically selected during the week? Was the observation period uniform between different schools? Observing a class in school A in the morning on Tuesday PE class (9-11am) would result in significantly different behavior of the students in school B who were observed on Wednesday afternoon 2-4pm (their normal class). Some clarifications on this matter should be added.

Author Response: Given the volunteers’ varied availability to conduct the observations and the schools’ different and, at times, dynamic, schedules, it was impossible to adhere to a uniform observation schedule across all three schools. However, as we state in our response to the reviewer’s comment above, our aim was to combine the data we collected across all three schools to create a holistic (albeit initial) perspective of school-based implementation of Dubai 30x30. Considering the cumulative results from our observations, which spanned 15 classrooms in 12 days throughout the full month of Dubai 30x30, we were successful in ensuring we captured evidence of different periods of the school day (e.g., when students were in physical education class, in their homeroom classrooms, at lunch, inside, outside, etc.), as shown in our results related to Research Question 3. 

6. L487-488: Please note that the two schools involved are international schools with British or European children as majority. In the discussion, the authors should also address the limitation in the generalization of this study considering the convenience sampling being used and the selection of the participants.

Author Response: We have noted this limitation in our discussion section.

7. L514-518: the authors should further discuss to what extent the activities are different from their regular practice. The authors should also mention and discuss, what kind of activities were provided in the PE class, during breaks and before/after school. Were the activities also provided before Dubai 30x30 initiatives? If yes, does it mean schools were just do as their normal? Or is there any new activities that schools have done to participate in the Dubai 30x30 initiatives? (crosscheck the arguments with the students’ responses on motivation).

Author Response: Our interview and focus group data that we included in the manuscript provide all the details we have about the types of activities that were implemented during Dubai 30x30. We reported interview questions that we used (e.g., “What did your school do in the name of Dubai 30x30 this year?”), which focus directly on activities and experiences that were specifically implemented during Dubai 30x30. Therefore, from what can surmise from the data, the activities that participants described were mostly unique to Dubai 30x30 and were not merely part of routine practice at the schools.

8. L381: The findings showed an inconsistent result, between teachers’ vs students’ perspective regarding their participation. This should be discussed further; whether the teachers responded normatively as they are expected to show school participation?

Author Response: As we explained in our discussion, we feel these conflicting results are mostly likely due to occasions where staff provided activities, but students opted not to participate, possibly due to a lack of motivation or interest. We wish to reiterate that while school staff were asked in the survey to indicate the activities that were provided during Dubai 30x30, students were asked to indicate which activities they participated in. Thus, it is not that surprising that there are differences in the results for Research Questions 1 and 2.

Few minor points to help improve the readers’ understanding:

9. Tables should be self-explanatory, but I find it difficult to understand the tables presented. What is the unit of M in table 2? Please clarify whether it’s number of activities, or minutes? Was it times per week? Or per month? Please also clarify how many schools are involved.

Author Response: We have added this information to the tables for clarification. The information can also be found in the methods section.

10. Please also clarify the unit of table 4. See previous comment for table 2: 45 students from how many schools?

Author Response: We have added this information to the table for clarification. This information can also be found in the methods section.

11. Table 5: Relationship between Students’ PA Levels and School Context. The outcome of the study was not clear; why physical activity of the students was assessed on its association with the school contexts, but not on how it’s related to Dubai 30x30?

Author Response: This research question addressed the extent and level of physical activity students participated in from an observational (i.e., more objective) perspective. The intent here was to use these data to build on the survey, interview, and focus group data. As we were unable to observe all activities and events during schools’ implementation of Dubai 30x30, we felt combining observations with surveys, interviews, and focus groups would help to create a more comprehensive picture of the implementation experience. The OSRAC-E tool allowed us to see which contexts yielded the most physical activity among the students who were observed. The data from this tool helped us to better understand which school contexts seemed to get more and less attention as places to promote physical activity during Dubai 30x30.

12. State the unit of M in Table 6. What does the score mean?

Author Response: We have added this information to the table. It is also available in the methods section.

---

## [Decision Letter · Decision Letter 1]

24 Jan 2024

Whole-of-School Physical Activity Implementation in the Context of the Dubai Fitness Challenge

PONE-D-23-25904R1

Dear Dr. Webster,

We’re pleased to inform you that your manuscript has been judged scientifically suitable for publication and will be formally accepted for publication once it meets all outstanding technical requirements.

Kind regards,

Josephine N. Booth

Academic Editor

PLOS ONE

Additional Editor Comments (optional):

Thank you for addressing all of the suggestions for your manuscript. I am pleased to be able to accept this for publication.

Reviewers' comments:

Reviewer's Responses to Questions

**Comments to the Author**

1. If the authors have adequately addressed your comments raised in a previous round of review and you feel that this manuscript is now acceptable for publication, you may indicate that here to bypass the “Comments to the Author” section, enter your conflict of interest statement in the “Confidential to Editor” section, and submit your "Accept" recommendation.

Reviewer #1: All comments have been addressed

Reviewer #2: All comments have been addressed

2. Is the manuscript technically sound, and do the data support the conclusions?

Reviewer #1: Yes

Reviewer #2: Yes

3. Has the statistical analysis been performed appropriately and rigorously? 

Reviewer #1: N/A

Reviewer #2: Yes

4. Have the authors made all data underlying the findings in their manuscript fully available?

Reviewer #1: Yes

Reviewer #2: Yes

5. Is the manuscript presented in an intelligible fashion and written in standard English?

Reviewer #1: Yes

Reviewer #2: Yes

6. Review Comments to the Author

Reviewer #1: The author(s) have addressed concerns that were raised in the previous version of the manuscript. There is also a good attempt to indicate how the Physical Activity approach guided the authors in the selection of the research design, sampling, and research methods. This contributes towards helping the reader to look at the manuscript as a cohesive unit from the introduction to the conclusion. There is also a detailed explanation of the justification for the use of convenient parallel mixed-methods research design. Ethical issues have been well outlined and the pseudonym has been used to hide the name of the ethics board. There is also an explanation of the steps that were taken to ensure that data generation was not compromised. However, there is a need to remove yellow colours in some of the sentences towards the end of the manuscript. They are found in sentences numbered 528, and 529 on page 30. Page 32, line 539. Lines 557, 558, 539, 682, 683 in page 34. Lines 698, 699, 700, 708, 709,110, 711, 712. There might be other pages where sentences are marked in yellow. I, therefore, suggest that the authors should go through the manuscript to verify the removal of the identified colours.

Reviewer #2: The authors have addressed all of the comments and suggestions. My main concern was the unclarity of the study design, including the approach and the sample (school and students) selection. While there are still some limitation of the study due to its design, the authors have clarified that the study only intended as a descriptive rather an analytic or evaluative one.

7. PLOS authors have the option to publish the peer review history of their article (what does this mean?). If published, this will include your full peer review and any attached files.

Reviewer #1: **Yes: **Layane Thomas Mabasa

Reviewer #2: **Yes: **Dyah Anantalia Widyastari

---

## [Editor Report · Acceptance letter]

2 Mar 2024

PONE-D-23-25904R1 

PLOS ONE

Dear Dr. Webster, 

I'm pleased to inform you that your manuscript has been deemed suitable for publication in PLOS ONE. Congratulations! Your manuscript is now being handed over to our production team.

Kind regards, 

on behalf of

Dr. Josephine N. Booth 

Academic Editor

PLOS ONE